# Immune Checkpoint Inhibitor-Related Cytopenias: About 68 Cases from the French Pharmacovigilance Database

**DOI:** 10.3390/cancers14205030

**Published:** 2022-10-14

**Authors:** Mickaël Martin, Hoan-My Nguyen, Clément Beuvon, Johana Bene, Pascale Palassin, Marina Atzenhoffer, Franck Rouby, Marion Sassier, Marie-Christine Pérault-Pochat, Pascal Roblot, Marion Allouchery, Mathieu Puyade

**Affiliations:** 1Service de Médecine Interne, Centre Hospitalier Universitaire de Poitiers, 86000 Poitiers, France; 2Faculté de Médecine et de Pharmacie, Université de Poitiers, 86000 Poitiers, France; 3Institut National de la Santé et de la Recherche Médicale U1313, Université de Poitiers, 86000 Poitiers, France; 4Centre Régional de Pharmacovigilance du Nord-Pas de Calais, Centre Hospitalier Universitaire de Lille, 59000 Lille, France; 5Département de Pharmacologie Médicale et Toxicologie, Centre Hospitalier Universitaire de Montpellier, 34295 Montpellier, France; 6Service Hospitalo, Universitaire de Pharmaco-Toxicologie, Hospices Civils de Lyon, 69424 Lyon, France; 7CRPV Marseille Provence Corse, Service Hospitalo, Universitaire de Pharmacologie Clinique et Pharmacovigilance, Assistance Publique Hôpitaux de Marseille, 13385 Marseille, France; 8Département de Pharmacologie, Centre Hospitalier Universitaire de Caen, 14033 Caen, France; 9Pharmacologie Clinique et Vigilances, Centre Hospitalier Universitaire de Poitiers, 86000 Poitiers, France; 10Laboratoire de Neurosciences Expérimentales et Cliniques, Institut National de la Santé et de la Recherche Médicale U1084, Université de Poitiers, 86000 Poitiers, France; 11Centre d’Investigation Clinique-1402, Centre Hospitalier Universitaire de Poitiers, 86000 Poitiers, France

**Keywords:** immune checkpoint inhibitor, autoimmune cytopenia, autoimmune hemolytic anemia, immune thrombocytopenia, neutropenia, pure red cell aplasia, aplastic anemia, immune-related adverse event

## Abstract

**Simple Summary:**

Data on immune checkpoint inhibitor (ICI)-related cytopenias are scarce. The aim of the study was to further characterize grade ≥ 2 ICI-related cytopenias using the French pharmacovigilance database. Immune thrombocytopenia and autoimmune hemolytic anemia were the most frequently reported ICI-related cytopenias (50.7% and 25.3%, respectively). Nearly half were grade ≥ 4, and 4.4% of patients died from cytopenia-related complications. Using the French pharmacovigilance database, this study provides a comprehensive analysis of ICI-related cytopenias that are rare but potentially life-threatening adverse drug reactions. Early recognition and timely initiation of appropriate treatment are key in their management in clinical practice.

**Abstract:**

Immune checkpoint inhibitor (ICI)-related cytopenias have been poorly described. This study aimed to further characterize ICI-related cytopenias, using the French pharmacovigilance database. All grade ≥ 2 hematological adverse drug reactions involving at least one ICI coded as suspected or interacting drug according to the World Health Organization criteria and reported up to 31 March 2022, were extracted from the French pharmacovigilance database. Patients were included if they experienced ICI-related grade ≥ 2 cytopenia. We included 68 patients (75 ICI-related cytopenias). Sixty-three percent were male, and the median age was 63.0 years. Seven patients (10.3%) had a previous history of autoimmune disease. Immune thrombocytopenia (ITP) and autoimmune hemolytic anemia (AIHA) were the most frequently reported (50.7% and 25.3%, respectively). The median time to onset of ICI-related cytopenias was 2 months. Nearly half were grade ≥ 4, and three patients died from bleeding complications of refractory ITP and from thromboembolic disease with active AIHA. Out of 61 evaluable responses, complete or partial remission was observed after conventional treatment in 72.1% of ICI-related cytopenias. Among the 10 patients with ICI resumption after grade ≥ 2 ICI-related cytopenia, three relapsed. ICI-related cytopenias are rare but potentially life-threatening. Further studies are needed to identify risk factors of ICI-related cytopenias.

## 1. Introduction

Immune checkpoint inhibitors (ICIs) are nowadays the standard of care in the treatment of different types of cancers [1]. ICIs promote antitumor immune response through inhibition of cytotoxic T-lymphocyte-associated-4 (CTLA-4) (ipilimumab, tremelimumab), programmed cell death protein 1 (PD-1) (nivolumab, pembrolizumab and cemiplimab) or PD ligand 1 (PD-L1) (atezolizumab, durvalumab and avelumab) [2,3,4]. Despite often durable responses to ICI treatment, its use has been associated with a broad spectrum of immune-related adverse events (irAEs), due to the ICI-induced T cell cytotoxicity and B-cell synthesis of autoantibodies, possibly leading to ICI-related cytopenias [2]. However, precise underlying mechanisms remain unknown. IrAEs could be severe and affect various organs or system [5]. With the rapid and broad expansion of ICIs in clinical practice, a concern emerged about the risk of hematological irAEs in patients receiving ICI treatment. In fact, ICI-related cytopenias have been poorly described because of low incidence and possibly lack of recognition. Indeed, diagnosis of ICI-related cytopenias remains challenging, especially in patients receiving ICIs in combination with cytotoxic chemotherapy or radiotherapy. The largest published series of hematological irAEs arising from ICIs to date did not provide any clinical data regarding diagnosis criteria or management [6,7,8], and the conclusions were consequently limited. Using the French pharmacovigilance database, this study aimed to further characterize ICI-related cytopenias after ICI treatment.

## 2. Materials and Methods

### 2.1. Data Source

Data were extracted from the French pharmacovigilance database (accessed on 31 March 2022) that includes all adverse drug reactions spontaneously reported by healthcare specialists from the 30 French Regional Pharmacovigilance Centers and reviewed by pharmacologists. The French pharmacovigilance database is administered by the French Medicine Agency. Causality is assessed according to the French method for causality assessment of adverse drug reactions published by Bégaud et al. [9]. The Medical Dictionary for Regulatory Activities is used to report adverse drug reactions in the database [10].

### 2.2. Data Extraction and Selection

Using MedDRA terms ‘Blood and lymphatic disorders’ and ‘Investigations’ (System Organ Class Level), all grade ≥ 2 hematological adverse drug reactions involving at least one of the ICIs (including nivolumab, pembrolizumab, cemiplimab, ipilimumab, tremelimumab, atezolizumab, durvalumab and avelumab) coded as suspected or interacting drugs according to the World Health Organization criteria and reported up to 31 March 2022, were extracted from the French pharmacovigilance database. Patients were eligible if they experienced grade ≥ 2 cytopenia after ICI treatment. Diagnosis and response criteria for each ICI-related cytopenia are provided in Methods S1 [11,12,13,14,15,16]. Patients were excluded if cytopenia was probably not caused by ICIs (i.e., chemotherapy-induced cytopenia, systemic lupus erythematosus [17], chronic lymphocytic leukemia [18] or ongoing infection [13,14,19,20,21,22]). Doubtful cases were reviewed by a board of experts in hematology, autoimmune diseases, and pharmacovigilance (MA, MP and MM). ICI resumption was defined, as previously published [23], as ICI discontinuation for a period at least equal to twice the duration of a cycle, with subsequent ICI restarting.

### 2.3. Data Collection

Patient general characteristics, including age, sex, body mass index and characteristics of the ICIs (type, indication, number of courses, resumption) were recorded. For immune-related cytopenias, the following characteristics were collected: type, time to occurrence, grade of severity according to the National Cancer Institute Common Terminology Criteria for Adverse Events v5.0, as well as their treatment(s) and outcomes.

### 2.4. Statistical Analysis

Quantitative variables were described by median and interquartile range (IQR) or mean and standard deviation (SD). Qualitative variables were described by numbers and proportions. All statistical analyses were performed using SAS software (v9.4, SAS Institute, Cary, NC, USA).

### 2.5. Ethics Approval and Consent

In accordance with French regulations, this study did not require ethical review. The French pharmacovigilance database was registered with the French data protection agency (Commission Nationale de l’Informatique et des Libertés) (deliberation No. 2014-302, 10 July 2014). Informed consent was not required as it involved an existing anonymized database.

## 3. Results

### 3.1. Patient and ICI Treatment Characteristics

We included 68 patients with a total of 75 ICI-related cytopenias (Figure 1). Patient characteristics are detailed in Table 1. Sixty-three percent were men, and the median age was 63.0 (58.0–70.5) years. Seven out of 68 (10.3%) patients had a history of autoimmune disease. The most common malignancies were lung cancer (47.1%) and melanoma (32.4%). Regarding the types of ICIs, 80.9% of patients were receiving an anti-PD-1 agent, 8.8% an anti-CTLA-4 and anti-PD-1 combination, 7.4% an anti-PD-L1 agent, and 2.9% an anti-CTLA-4 agent. ICI treatment was prescribed as first-line treatment and combined with chemotherapy in 47.2% and 7.5%, respectively. Three patients (4.4%) died from ICI-related cytopenias: two due to bleeding complications of refractory immune thrombocytopenic purpura (ITP) and one due to pulmonary embolism and ischemic stroke related to patent foramen ovale in a context of active autoimmune hemolytic anemia (AIHA).

### 3.2. Characteristics of the ICI-Related Cytopenias

A total of 75 ICI-related cytopenias after ICI treatment were reported for the 68 patients; seven (10.3%) patients experienced two ICI-related cytopenias (4 with ITP and AIHA, 2 with ITP and autoimmune neutropenia (AIN), and 1 with AIHA and pure red cell aplasia (PRCA)). ITP and AIHA were the most common ICI-related cytopenias (50.7% and 25.3%, respectively) (Table 2). The median number of cycles prior to the onset of immune-related cytopenia was 3.0 (2.0–6.0) with a maximum of 26 cycles (32 months). Aplastic anemia (AA) occurred later after ICI treatment (7.0 cycles/4.7 months) than other immune-related cytopenias. Grade 3 toxicities occurred in 39.2%, grade 4 in 44.6% and grade 5 in 4.0% of the patients. Detailed characteristics of ICI-related cytopenias are provided in Appendix A.

Considering treatment of ICI-related cytopenias (*n* = 72), 56 (77.8%) patients received glucocorticoids, 12 (16.7%) intravenous immunoglobulins and three (4.2%) immunosuppressive drugs as first-line treatment. Three (4.2%) patients required second-line immunosuppressant treatments, including cyclosporine for AA, rituximab for AIHA and cyclophosphamide for PRCA (*n* = 1 for each). Thrombopoietin agonists were used in two patients with ITP as second- or later-line treatment and in one patient with AA. As part of supportive care, five (6.9%) patients were given granulocyte colony-stimulating factor and two (2.8%) erythropoiesis-stimulating agents. Out of 61 evaluable responses, complete and partial remission was observed in 30 (49.2%) and 14 (23.0%) of the ICI-related cytopenias, respectively. A relapse of ICI-related cytopenias other than ICI resumption was mentioned in six (11.0%) cases. Ten patients resumed ICI treatment, leading to the relapse of three ICI-related cytopenias (2 ITP and 1 AIN).

## 4. Discussion

As ICI-related cytopenias remained poorly described due to low incidence and confounding factors, this study aimed to provide complementary data on ICI-related cytopenias in real-life practice using the French pharmacovigilance database. 

Among the wide variety of ICI-related cytopenias in this study, ITP and AIHA were the most frequently reported (50.7% and 25.3%, respectively). ICI-related cytopenias overlapped in only seven patients (10.3%). These findings are consistent with those reported in a case-series from the World Health Organization pharmacovigilance database VigiBase^®^ [7], but with higher rates of ITP than AIHA in our study. However, due to the absence of diagnosis criteria in the adverse drug reactions reports, definitive causality of ICIs could not be established by the authors based on VigiBase^®^ reporting.

The median time to onset of ICI-related cytopenias of 8 weeks is similar to that reported in the literature, ranging from 6 to 10.1 weeks [6,8,24]. Indeed, our data suggest that AA may occur later (4.7 months) than ITP, AIHA, PRCA or AIN (1 to 2.4 months). However, the low number of AAs (*n* = 2) does not allow any definitive conclusion.

Most of the patients (80.0%) experienced ICI-related cytopenias after anti-PD-1 treatment. Higher rates of ICI-related cytopenias were previously reported with anti-PD-1 monotherapy than with anti-CTLA-4 monotherapy [25,26] or ICI combination [26]. Michot et al. reported similar rates of hematological irAEs after anti-PD-(L)1 monotherapy [25]. In addition, anti-PD-(L)1 monotherapy was implicated in 62.0% of hematologic irAEs after ICI treatment in VigiBase^®^ [7]. Conversely, ITP incidence appeared highest among those treated with ICI combination approaches in two retrospective studies [27,28]. Whether ICI combination or type of ICI could influence time to onset of hematological irAEs also remains unknown. Hematological irAEs may occur earlier after the onset of ICI combination versus ICI monotherapy [6]. Indeed, anti-CTLA-4 based therapy rather than anti-PD-(L)1 monotherapy was more frequently associated with early-onset hematological irAEs [7]. 

ICI-related cytopenias are rare but potentially life-threatening adverse drug reactions. Almost half (48.6%) were grade ≥4, and three patients (4.4%) died from complications of ITP or AIHA. Grade ≥4 hematological irAEs were reported in 77% of patients, and two patients (6%) in a previous descriptive study died from febrile ICI-related neutropenia [24]. By contrast, in a more recent observational study, Kramer et al. reported lower rates of grade ≥4 hematological irAEs (28.0%) and only one death (2%) related to ICI-associated cytopenia [6]. All in all, mortality rates due to ICI-related cytopenias vary widely across studies (2–16%) [6,7,24,25,26]. These discrepancies could be related to the nature of data collection (underreporting of lower grade irAEs in register-based or pharmacovigilance studies) and to improved detection and management of ICI-related cytopenias over time.

The overall complete response rate of ICI-related cytopenia in our study was 50%. A recent literature review showed a high rate (79%) of full resolution of ICI-related cytopenias in a longer term perspective, raising questions on ICI resumption as an option for these patients [25]. In our study, among the 10 rechallenged patients, three experienced recurrence of the same ICI-related cytopenia, which is slightly lower than the 40–57% reported in the literature [6,24,26]. Interestingly, the recurrence rate of ICI-related cytopenias in our study was quite similar to the recurrence rate observed for irAEs of all types [23]. ICI resumption after grade ≥2 ICI-related cytopenias needs to be discussed in a multidisciplinary team meeting, considering potential efficacy of resumption, patient comorbidities and risk of recurrence. Haanen et al. proposed an algorithm for ICI resumption after hematological irAEs, but it only concerned ITP or AIHA occurring after ICI treatment [29].

The specific mechanisms of ICI-related cytopenias remain unknown. PD-1/PD-L1 axis may be crucial in prevention of immune-mediated damage of the hematopoietic niche and partially explain why ICI-related cytopenias are more frequent with anti-PD(L)-1 agents [30,31,32]. By blocking inhibitory signals to cytotoxic and helper T lymphocytes and suppressing Tregs activation (anti-CTLA-4) while blocking inhibitory signals to B lymphocytes, NK cells, and macrophages (anti-PD(L)-1), ICIs unleash T-lymphocyte/NK cytotoxicity and B-cell synthesis of autoantibodies, possibly leading to ICI-related cytopenias (anemia, thrombocytopenia, neutropenia or bone marrow failure) [33]. In our study, 10.3% of patients had a previous history of autoimmune disease, in line with the 14% reported by Gnanapandithan et al. [26]. Several factors have been suggested as increasing the risk of overall ICI toxicity without any apparent effect on efficacy (i.e., HLA-DR4/DRB1*04:05/DRB1*11:01, history of autoimmune diseases, baseline autoantibody or cytokine levels, and the ratio of neutrophils or platelets to lymphocytes) [34]. Further studies are needed to identify patients at risk of ICI-related cytopenias.

Our study has several strengths. First, all cases of ICI-related cytopenias were reviewed by a board of three experts in hematology, autoimmune diseases, and pharmacovigilance, based on well-defined diagnosis criteria. Moreover, access to clinical data allowed us not only to confirm immune-related cytopenias, but also to assess ICI causality and exclusion of potential confounders. Second, the real-life study design, including a wide range of ICI regimens and cancer types, and its large scale, limited selection bias.

The main limitation of our study is intrinsic to the pharmacovigilance system itself. Being based on spontaneous reporting of ADRs by healthcare professionals, under-reporting of less severe immune-related cytopenias cannot be ruled out [35,36]. Selective reporting is also a possibility, with fatal cases more likely to be reported than non-fatal ones. In addition, as pharmacovigilance databases do not capture the number of patients exposed to ICIs, the frequency of ICI-related cytopenias could not be assessed. Given the retrospective nature of data collection, several data, such as clinical and biological work-up in the context of ICI-related cytopenias, were missing, leading to the exclusion of potential cases.

## 5. Conclusions

ICI-related cytopenias, which are mainly represented by ITP and AIHA, are rare but potentially life-threatening. Monitoring of complete blood count appears to be the easiest way to achieve early detection of these irAEs. As the diagnosis of ICI-related cytopenias remains challenging, referral to an autoimmune cytopenia specialist should be required in case of suspicion of ICI-related cytopenia. Due to the bias inherently associated with pharmacovigilance studies, prospective studies are needed to determine risk factors for ICI-related cytopenias.

## Figures and Tables

**Figure 1 cancers-14-05030-f001:**
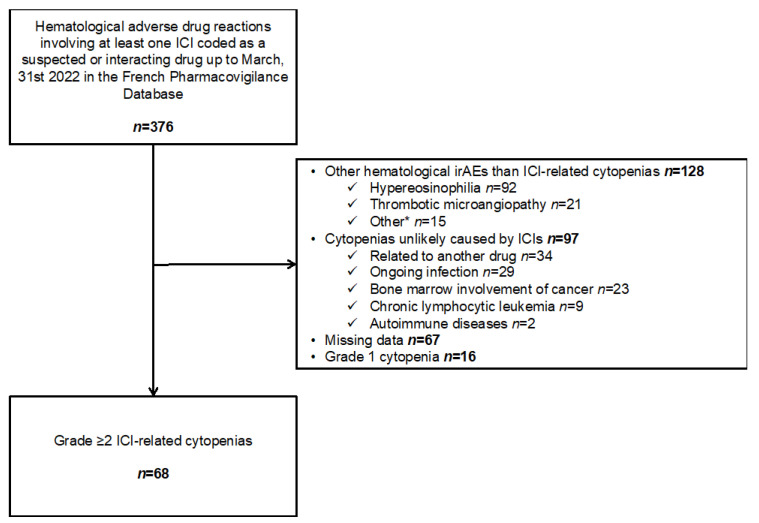
Flow chart. * Lymphopenia (*n* = 8), bleeding event without cytopenia (*n* = 4), hemophagocytic lymphohistiocytosis (*n* = 1), lymph node enlargement (*n* = 1), no cytopenia after experts’ review (*n* = 1). ICI: Immune checkpoint inhibitor, irAEs: Immune-related adverse events.

**Table 1 cancers-14-05030-t001:** Characteristics of the study population.

	*N* = 68	
Age, years, median (IQR)	63.0	(58.0–70.5)
Male gender	43	(63.2)
Body mass index, kg/m^2^, median (IQR) (*n* = 36)	24.8	(21.6–30.3)
History of hematologic malignancies ^†^	5	(7.4)
Monoclonal gammopathy of undetermined significance	1	(1.5)
Myelodysplastic syndrome	2	(2.9)
Multiple myeloma	1	(1.5)
Diffuse large B cell lymphoma in remission	1	(1.5)
History of cancer	6	(8.8)
History of autoimmune disease (*n* = 66)	7	(10.3)
Cured ITP	2	(3.0)
Crohn’s disease	1	(1.5)
Hypothyroidism	2	(3.0)
Sarcoidosis	1	(1.5)
Overlap Sjogren’s syndrome/Behçet’s disease	1	(1.5)
ICI-treated cancer type		
Lung cancer	32	(47.1)
Melanoma	22	(32.4)
Renal cell carcinoma	4	(5.8)
Head and neck squamous cell carcinoma	3	(4.4)
Urinary cancer	2	(2.9)
Other	5	(7.4)
ICI as first-line treatment (*n* = 53)	25	(47.2)
ICI		
Anti-PD-1	55	(80.9)
Pembrolizumab	32	(47.1)
Nivolumab	23	(33.8)
Anti-PD-1 + anti-CTLA-4 (nivolumab and ipilimumab)	6	(8.8)
Anti-PD-L1	5	(7.4)
Atezolizumab	3	(4.4)
Durvalumab	1	(1.5)
Avelumab	1	(1.5)
Anti-CTLA-4 (ipilimumab)	2	(2.9)
Concomitant chemotherapy (*n* = 67)	5	(7.4)
Death from immune-related cytopenia	3	(4.4)
ICI resumption after immune-related cytopenia ^‡^	10	(14.7)

Data are expressed as *n* (%) unless otherwise stated. ^†^ Other than ICI indication. ^‡^ ICI discontinuation for a period at least equal to twice the duration of a cycle, with subsequent ICI restarting. CTLA-4: Cytotoxic T-lymphocyte antigen-4, ICI: Immune checkpoint inhibitor, IQR: Interquartile range, ITP: Immune thrombocytopenic purpura, PD-1: Programmed cell death 1, PD-L1: Programmed cell death ligand.

**Table 2 cancers-14-05030-t002:** Characteristics of ICI-related cytopenias.

	All *N* = 75		ITP *N* = 38		AIHA *N* = 19		AIN *N* = 10 ^†^		PRCA *N* = 6		AA *N* = 2	
ICI												
Anti-PD-1	60	(80.0)	32	(84.2)	13	(68.4)	9	(90.0)	5	(83.7)	1	(50.0)
Anti-PD-L1	6	(8.0)	4	(10.5)	2	(10.5)	0	-	0	-	0	-
Anti-CTLA-4	2	(2.7)	1	(2.6)	1	(5.2)	0	-	0		0	-
Anti-PD-1 + anti-CTLA-4	7	(9.3)	1	(2.6)	3	(15.8)	1	(10.0)	1	(16.7)	1	(50.0)
Time to immune-related cytopenia, months, median (IQR)	2.0	(1.0–4.5)	1.8	(1.0–4.8)	1.4	(1.0–2.8)	2.4	(1.2–4.5)	2.1	(1.4–2.8)	4.7	NA
Time to immune-related cytopenia, treatment cycles, median (IQR)	3.0	(2.0–6.0)	3	(1.0–5.0)	2.0	(1.0–4.0)	3.5	(2.0–6.0)	3.0	(2.0–7.0)	7.0	NA
Maximum severity (*n* = 74)												
Grade 2	9/74	(12.2)	3/38	(7.9)	6/18	(33.3)	0/10	-	0/6	-	0/2	-
Grade 3	29/74	(39.2)	9/38	(23.6)	9/18	(50.0)	5/10	(50.0)	5/6	(83.3)	0/2	-
Grade 4	33/74	(44.6)	24/38	(63.2)	2/18	(11.1)	5/10	(50.0)	1/6	(16.7)	2/2	(100)
Grade 5	3/74	(4.0)	2/38	(5.3)	1/18	(5.6)	0/10	-	0/6	-	0/2	-
Treatment of immune-related cytopenia (*n* = 72)												
Number of lines, median (IQR)	1.0	(1.0–1.0)	1.0	(1.0–1.0)	1.0	(1.0–1.0)	1.0	(1.0–1.0)	1.5	(1.0–2.0)	1.0	(1.0–1.0)
Glucocorticoids,	56/72	(77.8)	32/36	(88.9)	16/19	(84.2)	2/10	(20.0)	5/5	(100.0)	1/2	(50.0)
Intravenous immunoglobulins	12/72	(16.7)	9/36	(25.0)	0/19	-	1/10	(10.0)	2/5	(40.0)	0/2	-
G-CSF	5/72	(6.9)	0/36	-	0/19	-	5/10	(50.0)	0/5	-	0/2	-
Thrombopoietin agonist	3/72	(4.2)	2/36	(5.6)	0/19	-	0/10	-	0/5	-	1/2	(50.0)
Cyclosporine	3/72	(4.2)	0/36	-	1/19	(5.3)	0/10	-	1/5	(20.0)	1/2	(50.0)
ESA	2/72	(2.8)	0/36	-	0/19	-	0/10	-	2/5	(40.0)	0/2	-
Rituximab	2/72	(2.8)	0/36	-	2/19	(10.5)	0/10	-	0/5	-	0/2	-
Cyclophosphamide	1/72	(1.4)	0/36	-	0/19	-	0/10	-	1/5	(20.0)	0/2	-
Treatment response of immune-related cytopenia (*n* = 61)												
Complete	30	(49.2)	16/32	(50.0)	6/14	(42.9)	7/9	(77.8)	1/5	(20.0)	0/2	-
Partial	14	(23.0)	8/32	(25.0)	4/14	(28.6)	0/9	-	1/5	(20.0)	1/2	(50.0)
Stable disease	17	(27.9)	8/32	(25.0)	4/14	(28.6)	2/9	(22.2)	3/5	(60.0)	0/2	-
ICI resumption	10	(13.3)	5	(13.2)	2	(10.5)	3	(30.0)	0	-	0	-
Relapse of immune cytopenia after ICI resumption	3	(33.3)	2	(40.0)	0	-	1	(33.3)	0	-	0	-

Data are expressed as *n* (%) unless otherwise stated. ^†^ 2 AINs were not treated. AA: Aplastic anemia, AIHA: Autoimmune hemolytic anemia, AIN: Autoimmune neutropenia, CTLA-4: Cytotoxic T-lymphocyte antigen-4, ESA: Erythropoiesis stimulating agent, G-CSF: Granulocyte Colony Stimulating Factor, ICI: Immune checkpoint inhibitor, IQR: Interquartile range, ITP: Immune thrombocytopenia purpura, PD-1: Programmed cell death 1, PD-L1: Programmed cell death ligand 1, PRCA: Pure red cell aplasia.

## Data Availability

The data that support the findings of this study are available from the corresponding author upon reasonable request. The data are not publicly available due to privacy or ethical restrictions.

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
