# Peer review of "Immune Checkpoint Inhibitor-Related Cytopenias: About 68 Cases from the French Pharmacovigilance Database"

_cancers, 2022, doi:10.3390/cancers14205030_

Round 1

Reviewer 1 Report

The manuscript retrospectively describes the most common adverse effects of ICI. It is a pragmatic, concise report.

Suggestions:

1. The flow of reading is hampered by the abbreviations: Reduce them or add a list of abbreviations.

2. The (unclear) mechanisms of action of ICI is summarized under "discussion" (lines 212-225), but should also be shorthly mentioned in the "introduction". The same should be noted concerning the (unknown) number of patients with lower scores of the frequency of ICI-related cytopenias under "introduction".

Minor points:

Line 22: Omit "simple", say instead "Short"

Line 25: Incomplete sentence, add ... reported adverse events.

Line 138: Say AIHA (twice) instead of "AHIA"?

Table 1, last line and text: What do you mean with "resumption": relapse, recovery or re-occurance: clarify 

Author Response

Poitiers, October 4th, 2022

The manuscript retrospectively describes the most common adverse effects of ICI. It is a pragmatic, concise report.

We would like to thank the reviewer for his/her valuable comments.

The flow of reading is hampered by the abbreviations: Reduce them or add a list of abbreviations.

We have reduced the number of abbreviations in the text and in the tables. We have also added a list of abbreviations at the end of the manuscript. We have kept abbreviations for the different ICI-related cytopenias because they are commonly used in the literature.

The (unclear) mechanisms of action of ICI is summarized under "discussion" (lines 212-225), but should also be shortly mentioned in the "introduction". The same should be noted concerning the (unknown) number of patients with lower scores of the frequency of ICI-related cytopenias under "introduction".

We have modified the following sentence in the introduction part: "Despite often durable responses to ICI treatment, its use has been associated with a broad spectrum of immune-related adverse events (irAEs), due to the ICI-induced T cell cyto-toxicity and B-cell synthesis of autoantibodies, possibly leading to ICI-related cytopenias [2]. However, precise underlaying mechanisms remain unknownAlthough precise underlying mechanisms remain unknown” (lines 71-73).

The discussion part has been modified according to the comment of the reviewer (lines 188-191): “As ICI-related cytopenias remained poorly described due to low incidence and con-founding factors, this study aimed to provide complementary data on ICI-related cytopenias in real-life practice using the French pharmacovigilance database.”(lines 182-184)

Minor points:

Line 22: Omit "simple", say instead "Short"

The manuscript has been modified according to the comment of the reviewer (line 36).

Line 25: Incomplete sentence, add ... reported adverse events.

The sentence has been completed: “Immune thrombocytopenia and autoimmune hemolytic anemia were the most frequently reported ICI-related cytopenias (50.7% and 25.3% respectively)”(line 39).

Line 138: Say AIHA (twice) instead of "AHIA"?

The manuscript has been modified according to the comment of the reviewer (line 156). 

Table 1, last line and text: What do you mean with "resumption": relapse, recovery or re-occurrence: clarify 

ICI resumption means ICI restarting after ICI interruption (e.g. for irAEs). We have modified the manuscript (line 111): "ICI resumption was defined, as previously published [23], as ICI discontinuation for a period at least equal to twice the duration of a cycle, with subsequent ICI restarting.". We have also added a footnote on table 1 to define ICI resumption according to the text ("‡ICI discontinuation for a period at least equal to twice the duration of a cycle, with sub-sequent ICI restarting.") (lines 148-149).

Reviewer 2 Report

I find this paper well written and of great interest for hematologists dealing with this kind of complications. Just two minor ameliorative notes:

- Supplementary material, Appendix A1: in the diagnosis of AIHA, it is mentioned the direct "agglutinin test", whereas the correct form is direct ANTIGLOBULIN test (DAT)

- could you better characterize patients with AIHA and thromboembolic events? Which ICI had been used? It is in fact known that active hemolysis is associated to thrombosis, but also some ICI (e.g., nivolumab) are. I think it is worthy to discuss, even if only three patients experienced this complication

Author Response

I find this paper well written and of great interest for hematologists dealing with this kind of complications.

We would like to thank the reviewer for his interest and his/her compliments.

Just two minor ameliorative notes:

Supplementary material, Appendix A1: in the diagnosis of AIHA, it is mentioned the direct "agglutinin test", whereas the correct form is direct ANTIGLOBULIN test (DAT)

The supplementary material has been modified according to the comment of the reviewer.

Could you better characterize patients with AIHA and thromboembolic events? Which ICI had been used? It is in fact known that active hemolysis is associated to thrombosis, but also some ICI (e.g., nivolumab) are. I think it is worthy to discuss, even if only three patients experienced this complication

In fact, only one patient died from thromboembolic disease. We have therefore replaced the sentence and added some details: “Three patients (4.4%) died from ICI-related cytopenias: two due to bleeding complications of refractory immune thrombocytopenic purpura (ITP) and one due to pulmonary embolism and ischemic stroke related to patent foramen ovale in a context of active autoimmune hemolytic anemia (AIHA).” (lines 138-141).

Reviewer 3 Report

Good survey of a difficult problem.  Also nice to see outcomes although small in number.

Author Response

Good survey of a difficult problem.  Also nice to see outcomes although small in number.

We would like to thank the reviewer for his/her compliments.